# Mixture of Lookup Experts

**Shibo Jie**[1]   **Yehui Tang**[2]   **Kai Han**[2]   **Yitong Li**[3]   **Duyu Tang**[3]   **Zhi-Hong Deng**[1]   **Yunhe Wang**[2]

## Abstract

Mixture-of-Experts (MoE) activates only a subset of experts during inference, allowing the model to maintain low inference FLOPs and latency even as the parameter count scales up. However, since MoE dynamically selects the experts, all the experts need to be loaded into VRAM. Their large parameter size still limits deployment, and offloading, which load experts into VRAM only when needed, significantly increase inference latency. To address this, we propose Mixture of Lookup Experts (MoLE), a new MoE architecture that is efficient in both communication and VRAM usage. In MoLE, the experts are Feed-Forward Networks (FFNs) during training, taking the output of the embedding layer as input. Before inference, these experts can be reparameterized as lookup tables (LUTs) that retrieves expert outputs based on input ids, and offloaded to storage devices. Therefore, we do not need to perform expert computations during inference. Instead, we directly retrieve the expert's computation results based on input ids and load them into VRAM, and thus the resulting communication overhead is negligible. Experiments show that, with the same FLOPs and VRAM usage, MoLE achieves inference speeds comparable to dense models and significantly faster than MoE with experts offloading, while maintaining performance on par with MoE. Code: https://github.com/JieShibo/MoLE.

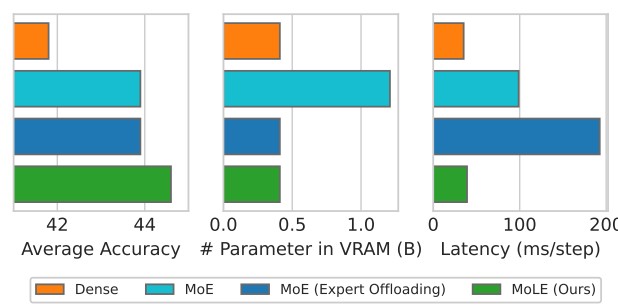

*Figure 1.* With the same 410M activated parameters, MoE outperforms the dense model in terms of performance, but it comes with significant VRAM usage. If experts are offloaded, inference latency will increase. Our MoLE maintains competitive performance without increasing the model's VRAM usage or decoding latency.

## 1. Introduction

Scaling laws indicate that, with sufficient data for training, the performance of large language models (LLMs) improves as the model size increases (Kaplan et al., 2020). However, larger LLMs also result in slower inference speeds, which can degrade the user experience. For this reason, the architecture of LLMs has increasingly focused on Mixture of Experts (MoE) (Jiang et al., 2024; Dai et al., 2024). MoE models use several Feed-Forward Networks (FFNs) as experts and employ a router to determine which subset of experts needs to be activated, rather than activating the entire model. This allows the model to maintain a large number of parameters while keeping the computational cost low.

Although MoE reduces the computational cost, the number of parameters does not decrease. This means that the VRAM requirements during inference remain unaffordable. For example, although the Mixtral-8×7B (Jiang et al., 2024) model only has 13B parameters activated at a time, its total parameter count reaches 46B, making it impossible to load into a single 80GB GPU with FP16. Existing methods (Eliseev & Mazur, 2023; Xue et al., 2024; Shen et al., 2022) reduce VRAM usage by offloading experts to larger storage devices (*e.g.*, CPU RAM, disk, or cloud storage), and loading the selected experts into VRAM at each inference step. However, there are two drawbacks to this approach: *i)* Since the selection of experts is dynamically

---

[1]State Key Laboratory of General Artificial Intelligence, School of Intelligence Science and Technology, Peking University [2]Huawei Noah's Ark Lab [3]Consumer Business Group, Huawei. Correspondence to: Yunhe Wang <yunhe.wang@huawei.com>, Zhi-Hong Deng <zh-deng@pku.edu.cn>, Yehui Tang <yehui.tang@huawei.com>.

determined by the router, we must load different experts into VRAM at each inference step. Frequent transfer of large numbers of parameters can significantly increase inference latency, as in Figure 1. *ii)* Since different samples select different experts within a single step, loading only a subset of experts may not meet the needs of batched generation.

To address the issues mentioned above, we propose Mixture of Lookup Experts (MoLE), a new LLM architecture. MoLE has different structures in training and inference. During training, MoLE is similar to MoE, with a router and several experts. However, unlike MoE, where experts take intermediate features as input, MoLE's experts are fed with embedding tokens (*i.e.*, the output of the embedding layer) instead. Additionally, MoLE allows all experts to be activated simultaneously. After training, MoLE is not directly used for inference but undergoes a series of re-parameterizations. Since the output of the embedding layer is fixed for specific input ids, the inputs to the experts have only a limited number of choices, equal to the model's vocabulary size. Therefore, for each token in the embedding layer, we pre-compute the outputs corresponding to all experts, creating lookup tables (LUTs) that replaces the original experts.

During inference, MoLE demonstrates several advantages:

- **Computation-free experts.** The experts are re-parameterized from FFNs into LUTs, eliminating the need for any computation. Each expert only requires a single lookup operation.

- **Low VRAM overhead and communication latency.** Although the size of the LUT is much larger than the model itself, it can be entirely offloaded to storage devices. During inference, since the output of each expert is the same number of tokens as the input, the communication required to load the lookup results into VRAM is negligible, thereby avoiding increased inference latency.

- **Batch-generation friendly.** Traditional expert offloading methods introduce additional VRAM usage and communication latency during batched generation because different samples in a batch may select different experts. MoLE only transfers the pre-computed expert outputs, making its communication overhead still negligible even during batched generation.

Through extensive experiments, we validated the effectiveness of MoLE at scales of 160M, 410M, and 1B parameters. As in Figure 1, with equivalent computational cost and VRAM usage, MoLE significantly outperforms dense models while maintaining the same inference speed. Compared to MoE with expert offloading, MoLE achieves better performance and with significantly faster inference speed.

## 2. Related Work

### 2.1. Mixture-of-Experts

The concept of MoE was initially introduced by Jacobs et al. (1991); Jordan & Jacobs (1994) and has been widely explored and developed through subsequent research (Collobert et al., 2002; Rasmussen & Ghahramani, 2001; Shahbaba & Neal, 2009; Eigen et al., 2014; Theis & Bethge, 2015; Deisenroth & Ng, 2015; Aljundi et al., 2017; Shazeer et al., 2017). MoE posits that different parts of the model, *i.e.*, the experts, focus on distinct tasks or encapsulate different kinds of knowledge. In this paradigm, only the experts relevant to a given input are activated, which allows the model to scale its capacity while keeping computational costs manageable and making full use of specialized knowledge across many experts. As the scale of LLMs increases, reducing computational overhead has become a key focus. This leads to its application in transformer-based LLMs (Lepikhin et al., 2021), making MoE a widely used architecture.

Recently, a series of industrial-scale large language models have been released, including Mixtral (Jiang et al., 2024) and DeepSeek-MoE (Dai et al., 2024). Some works also aim to improve the efficiency of MoE (Jin et al., 2025) or increase expert capacity (Yan et al., 2025) by modifying the model architecture.

### 2.2. Expert Offloading

Offloading techniques typically transfer a portion of the model parameters to CPU RAM or disk when GPU memory is insufficient. However, current mainstream offloading frameworks, such as Zero-Infinity (Rajbhandari et al., 2021), are designed for dense LLMs and load model parameters layer by layer on-demand. This approach overlooks the sparse activation property of MoE models, leading to the unnecessary loading of inactive experts.

Building on this, some studies have proposed expert offloading, a form of parameter offloading specifically designed for the sparse activation characteristic of MoE models (Eliseev & Mazur, 2023; Xue et al., 2024). These methods stores non-expert weights and a portion of the expert cache in VRAM, while the remaining experts are offloaded to CPU RAM or disk and loaded on-demand.

Despite being effective, existing expert offloading techniques still suffer from high latency. Subsequent research includes optimizing prefetching techniques and cache replacement strategies to accelerate inference speed (Shen et al., 2022), designing MoE architectures that are more friendly to prefetching (Hwang et al., 2024), or employing other model compression techniques to reduce prefetching latency (Yi et al., 2023).

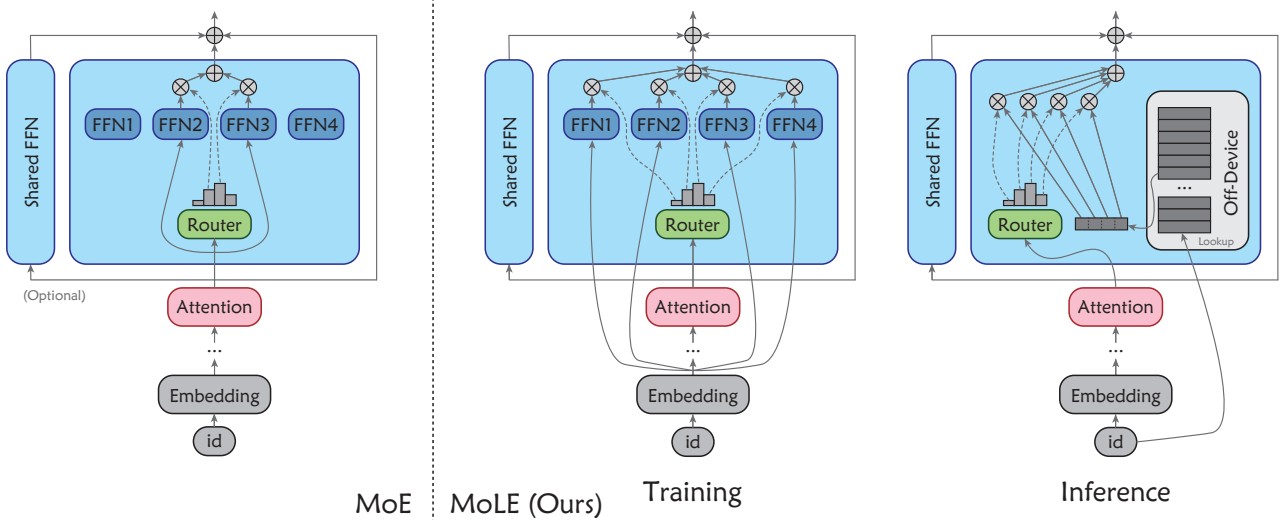

*Figure 2.* **Illustration of MoLE.** During training, MoLE differs from MoE in two key structural aspects: *i)* The routed experts in MoLE take embedding tokens as input. *ii)* All experts in MoLE are activated. During inference, the routed experts in MoLE are re-parameterized as zero-computation, offloaded LUTs. For simplicity, normalization layers and residual connections of attention layers are omitted.

## 3. Mixture of Lookup Experts

### 3.1. Preliminary

First, we briefly introduce the structure of MoE and the challenges it faces during inference.

For MoE, each expert is typically a FFN module. As illustrated in Figure 2 (left), a MoE layer contains $N$ routed experts, represented as $\{\text{FFN}_j\}_{j=1}^N$, and a linear router, denoted as $\{\boldsymbol{r}_j\}_{j=1}^N$. Some models may also introduce a shared expert $\text{FFN}_{shared}$ that is activated in all cases. Given input token $\boldsymbol{h} \in \mathbb{R}^d$, the output token $\boldsymbol{h}' \in \mathbb{R}^d$ of the MoE layer is computed as

$$\boldsymbol{G} = \texttt{ArgTopK}(\{\boldsymbol{h} \cdot \boldsymbol{r}_j\}_{j=1}^N) \tag{1}$$

$$\{g_j\}_{j \in \boldsymbol{G}} = \texttt{SoftMax}(\{\boldsymbol{h} \cdot \boldsymbol{r}_j\}_{j \in \boldsymbol{G}}) \tag{2}$$

$$\boldsymbol{h}' = \sum_{j \in \boldsymbol{G}} \big(g_j \text{FFN}_i(\boldsymbol{h})\big) + \text{FFN}_{shared}(\boldsymbol{h}) + \boldsymbol{h} \tag{3}$$

where $\boldsymbol{G}$ denotes the indexes of the activated experts, and $g_i$ denote the gate value for the $i$-th expert.

The computational efficiency of MoE lies in the fact that, in Eq. (3), only $k$ routed experts are involved in the computation. However, we cannot determine which experts need to participate until Eq. (1) is completed. This means that we either need to store all the experts in VRAM or temporarily load the required $k$ experts into VRAM after Eq. (1) is computed.

However, both of these solutions present deployment challenges. Taking Mixtral-8×7B as an example, it has 32

MoE layers, with 8 experts per layer, but only 2 experts are activated per token. Although only 13B parameters are activated per token, the total parameter count reaches up to 46B, requiring at least 92GB of VRAM for FP16 deployment.

If temporary loading is chosen, each expert is 176M in size, and loading the necessary experts for a single decoding step would require up to 11.3B of parameter transfer. If offloading to CPU VRAM is selected, using a GPU with PCIe 4.0×16 would still incur a transfer latency of 0.7s per step. Offloading to disk, on the other hand, results in an unacceptable transfer latency of over 10s per step. More importantly, since the selection of experts is dynamically determined by the router, the experts chosen for different samples are highly likely to differ when batch size > 1. This requires loading all the selected experts (may be all experts when batch size is large) into VRAM, which not only increases VRAM usage but also further exacerbates communication latency.

The reason experts need to be loaded into VRAM is that they participate in the computation, which relies on GPU. In other words, if the experts do not require computation, we do not need to load them, thereby avoiding significant communication overhead. To address this, we introduce MoLE, a new MoE architecture whose experts can be re-parameterized as computation-free LUTs in inference.

### 3.2. Training Phase

As illustrated in Figure 2, during training, MoLE and MoE have similar structures, including $N$ routed experts

*Table 1.* **Complexities of different architectures.** We report the statistics of a single FFN or MoE layer.

| Models | FLOPs | # Param in VRAM | # Param Offloaded | # Param Loaded per Token |
|---|---|---|---|---|
| Dense | $4dD_s$ | $2dD_s$ | 0 | 0 |
| MoE | $4d(kD_r + D_s)$ | $2d(ND_r + D_s)$ | 0 | 0 |
| MoE + Expert Offloading | $4d(kD_r + D_s)$ | $2d(kD_r + D_s)$ | $2dND_r$ | $2dkD_r$ (worst case) |
| MoLE + LUT Offloading | $4dD_s$ | $2dD_s$ | $dN|\mathcal{V}|$ | $dN$ |

$\{\text{FFN}_j\}_{j=1}^N$ and a linear router $\{r_j\}_{j=1}^N$. Specifically, MoLE also includes a shared expert $\text{FFN}_{shared}$, which is always activated for any input and does not receive weighting from the router.

Since the experts will be transformed into LUTs after training, MoLE differs from MoE in the following ways. First, LUT is computation-free, eliminating the need for sparse activation to reduce computational cost. Therefore, MoLE activates all experts, rather than just the top-$k$ experts. The computation for the router is as

$$\{g_j\}_{j=1}^N = \texttt{SoftMax}(\{\boldsymbol{h} \cdot \boldsymbol{r}_i\}_{i=1}^N) \tag{4}$$

Second, since the LUT is essentially a mapping between a finite set of input-output pairs, the key to re-parameterizing the experts into LUTs is ensuring that they only have a limited number of possible inputs. To this end, MoLE uses the output of the embedding layer, *i.e.*, the embedding tokens, as the input to the experts. After training, the embedding layer is only related to the input ids, which means that the inputs to the experts are limited to a finite set. The computation for the layer is as

$$\boldsymbol{h}' = \sum_{j=1}^N \big(g_j\text{FFN}_i(\boldsymbol{e})\big) + \text{FFN}_{shared}(\boldsymbol{h}) + \boldsymbol{h} \tag{5}$$

in which $\boldsymbol{e} = \text{Embedding}(i) \in \mathbb{R}^d$ is the embedding token, and $i$ deonotes the input id.

All experts are activated and receive gradients during training. Therefore, we do not need to add any auxiliary losses to prevent collapse or maintain training stability. MoLE is trained solely using the cross-entropy loss of language modeling just like a dense model.

### 3.3. Inference Phase

After training, MoLE can be directly used for inference like other LLMs. However, to further reduce VRAM overhead, we can re-parameterize the experts. For each possible input id $i$, we pre-compute the outputs of expert $\text{FFN}_j$ as

$$\boldsymbol{v}_j^i = \text{FFN}_j(\text{Embedding}(i)) \in \mathbb{R}^d \tag{6}$$

In practice, we only need to perform a single forward pass with the embedding layer's weights as the input to $\text{FFN}_j$,

which allows us to obtain $\boldsymbol{v}_j^i$ for all $i$. The items of the LUT at the $l$-th layer can be represented as

$$\text{LUT}_l = \{\{\boldsymbol{v}_j^i\}_{j=1}^N\}_{i=1}^{|\mathcal{V}|} \tag{7}$$

where $|\mathcal{V}|$ denotes the size of the vocabulary.

After re-parameterization, the LUT is offloaded to the storage device, and the computation of the MoLE layer can be represented as

$$\boldsymbol{h}' = \sum_{j=1}^N \big(g_j\boldsymbol{v}_j^i\big) + \text{FFN}_{shared}(\boldsymbol{h}) + \boldsymbol{h} \tag{8}$$

The inputs of the LUT in MoLE are the input ids, meaning that no context information is included. This is a trade-off to ensure that the experts can be re-parameterized, but it does not imply that the expert layers do not contribute to context-related knowledge. Firstly, the router and shared experts still take intermediate features as input, which means they can access contextual information. Secondly, the output of the expert layer is part of the input to subsequent attention layers, allowing the experts to influence the behavior of later attentions. This enables the experts to adjust how the model processes the same words in different contexts, thereby still enhancing the model's capacity.

### 3.4. Complexity Analysis

Consider an MoE layer with MLP-based FFNs as experts. Let the hidden layer dimension of the routed experts be $D_r$, and the hidden layer dimension of the shared experts be $D_s$. When a single token is used as input, the FLOPs for this MoE layer can be computed as

$$\text{FLOPs}_{\text{MoE}} = 4d(kD_r + D_s) \tag{9}$$

in which the router and normalization is neglected.

To save VRAM, we assume that all routed experts are offloaded, and then the offloaded parameter count is

$$\text{OffParam}_{\text{MoE}} = 2dND_r \tag{10}$$

In the worst case, during inference, we need to load the $k$ experts that the current token is routed to into VRAM. The number of per-step loaded parameter is

$$\text{LoadParam}_{\text{MoE}} = 2dkD_r \tag{11}$$

*Table 2.* **Model architectures.** We ensure a fair comparison by keeping the number of activated parameters the same in inference for the dense, MoE, and MoLE models.

| # Activated Param in Inference | Models | $L$ | $d$ | $D_s$ | $D_r$ | $N$ | $k$ | # Attention Heads | # Parameters in Training |
|---|---|---|---|---|---|---|---|---|---|
| 160M | Dense | 12 | 768 | 3072 | - | - | - | 12 | 0.16B |
| | MoE-10E | 12 | 768 | - | 1536 | 10 | 2 | 12 | 0.39B |
| | MoLE-4E | 12 | 768 | 3072 | 3072 | 4 | 4 | 12 | 0.39B |
| | MoE-34E | 12 | 768 | - | 1536 | 34 | 2 | 12 | 1.07B |
| | MoLE-16E | 12 | 768 | 3072 | 3072 | 16 | 16 | 12 | 1.07B |
| 410M | Dense | 24 | 1024 | 4096 | - | - | - | 16 | 0.41B |
| | MoE-10E | 24 | 1024 | - | 2048 | 10 | 2 | 16 | 1.21B |
| | MoLE-4E | 24 | 1024 | 4096 | 4096 | 4 | 4 | 16 | 1.21B |
| | MoE-34E | 24 | 1024 | - | 2048 | 34 | 2 | 16 | 3.63B |
| | MoLE-16E | 24 | 1024 | 4096 | 4096 | 16 | 16 | 16 | 3.63B |
| 1B | Dense | 16 | 2048 | 8192 | - | - | - | 8 | 1.01B |
| | MoE-10E | 16 | 2048 | - | 4096 | 10 | 2 | 8 | 3.16B |
| | MoLE-4E | 16 | 2048 | 8192 | 8192 | 4 | 4 | 8 | 3.16B |

For MoLE, since the experts are transformed into computation-free LUTs, its FLOPs can be computed as

$$\text{FLOPs}_{\text{MoLE}} = 4dD_s \quad (12)$$

The number of parameters contained in the offload LUT is

$$\text{OffParam}_{\text{MoLE}} = dN|\mathcal{V}| \quad (13)$$

In each inference step, since we only need to load all the $\boldsymbol{v}_j^i$ from Eq. (8) into VRAM, the amount of parameters loaded is only

$$\text{LoadParam}_{\text{MoLE}} = dN \quad (14)$$

We summarize all these comparisons in Table 1. Since $|\mathcal{V}|$ is typically on the order of tens of thousands, for example, $|\mathcal{V}| = 32k$ for Mixtral (Jiang et al., 2024) and $|\mathcal{V}| = 50k$ for Pythia (Biderman et al., 2023), and $D_r$ varies from thousands to tens of thousands depending on the model size, the number of offloaded parameters in MoE and MoLE will not differ by an order of magnitude. However, the number of parameters loaded per token in MoLE will be only a fraction — often hundreds or even thousands of times smaller — compared to the number of parameters loaded per token in MoE.

## 4. Experiments

### 4.1. Experimental Setup

**Model Architectures.** As shown in Table 2, we implement models with activation parameter counts of 160M, 410M, and 1B. For the dense model, we basically follow the Pythia (Biderman et al., 2023) setup. For the MoE model, we adopt a configuration similar to Mixtral, with no shared experts and activating the top-2 routed experts, since Muennighoff et al. (2024) suggest that shared experts lead to performance degradation. To ensure the number of activation parameter is the same as that of the dense model, the hidden dimension of the MoE FFNs is set to half the hidden dimension of the dense model's FFNs.

For the MoLE model, since routed experts have no computation during inference, we use FFNs identical to those of the dense model's FFNs as the shared experts to keep the same FLOPs as the dense model. For the routed experts, since their hidden dimension does not affect the model architecture during inference, we set it to be the same as the shared experts for simplicity. We implement MoE with both 10 and 34 experts and MoLE with both 4 and 16 experts for comparison.

**Data & Tokenizer.** We train all models on a 100B-token subset of the deduped Pile dataset (Gao et al., 2021), using the GPT-NeoX tokenizer employed by Pythia, with a vocabulary size of 50k.

**Hyper-Parameters.** We follow the learning rate settings used by Pythia, specifically $6.0 \times 10^{-4}$ for the models with 160M activated parameter, and $3.0 \times 10^{-4}$ for the models with 410M and 1B activated parameter. For the MoE model, the coefficients for the $z$-loss and load balance loss are set to 0.001 and 0.01, respectively, as suggested by Muennighoff et al. (2024).

**Benchmarks.** We use the lm-evaluation-harness package for evaluation. The benchmarks used include ARC-C (Clark et al., 2018), ARC-E (Clark et al., 2018), BoolQ (Clark et al., 2019), HellaSwag (Zellers et al., 2019), PIQA (Bisk et al., 2020), RACE (Lai et al., 2017), SIQA (Sap et al.,

*Table 3.* **Comparison of the complexities of different architectures.** MoE offloads all experts and MoLE offloads the LUT, ensuring that both models maintain the same VRAM usage as the dense model. "# Param Loaded per Token" considers the worst-case scenario, when the experts of MoE activated in the current step have no overlap with those from the last step.

| Models | # Param Offloaded | # Param Loaded per Token | ARC-C | ARC-E | BoolQ | HellaSwag | PIQA | RACE | SIQA | LAMBADA | AVG |
|---|---|---|---|---|---|---|---|---|---|---|---|
| *160M Activated & In-VRAM Parameters* | | | | | | | | | | | |
| Dense | 0B | 0M | 20.3 | 45.9 | 57.1 | 29.7 | 64.0 | 29.4 | 37.8 | 26.2 | 38.8 |
| MoE-10E | 0.3B | 57M | 21.7 | 49.5 | 51.6 | 32.0 | 66.8 | 30.6 | 39.1 | 31.0 | 40.3 |
| MoLE-4E | 1.8B | **0.037M** | 21.9 | 48.5 | 60.7 | 31.2 | 65.1 | 29.4 | 38.4 | 31.1 | **40.8** |
| MoE-34E | 1.0B | 57M | 20.5 | 50.0 | 57.5 | 34.5 | 67.3 | 28.6 | 39.9 | 36.4 | 41.8 |
| MoLE-16E | 7.4B | **0.15M** | 22.4 | 48.6 | 60.3 | 32.7 | 68.3 | 30.9 | 38.6 | 33.3 | **41.9** |
| *410M Activated & In-VRAM Parameters* | | | | | | | | | | | |
| Dense | 0B | 0M | 21.8 | 50.8 | 56.8 | 33.8 | 66.5 | 29.6 | 39.2 | 36.2 | 41.8 |
| MoE-10E | 1.0B | 201M | 24.1 | 53.5 | 54.5 | 37.1 | 69.0 | 30.8 | 40.8 | 41.5 | 43.9 |
| MoLE-4E | 4.9B | **0.098M** | 22.0 | 54.8 | 61.1 | 35.9 | 69.6 | 30.9 | 40.1 | 42.2 | **44.6** |
| MoE-34E | 3.4B | 201M | 25.0 | 57.0 | 59.7 | 39.9 | 71.5 | 32.3 | 40.4 | 47.1 | **46.6** |
| MoLE-16E | 19.7B | **0.39M** | 23.6 | 57.0 | 60.9 | 37.6 | 70.8 | 32.0 | 40.2 | 43.5 | 45.7 |
| *1B Activated & In-VRAM Parameters* | | | | | | | | | | | |
| Dense | 0B | 0M | 24.1 | 56.9 | 52.8 | 37.6 | 69.5 | 31.6 | 39.1 | 43.1 | 44.3 |
| MoE-10E | 2.7B | 537M | 25.9 | 57.8 | 53.8 | 40.7 | 72.0 | 33.6 | 41.3 | 48.0 | 46.6 |
| MoLE-4E | 6.6B | **0.26M** | 25.5 | 58.8 | 61.7 | 39.8 | 71.7 | 32.1 | 40.9 | 48.3 | **47.4** |

*Table 4.* **Ablation study on training loss.** The base model is MoLE-16E with 160M activated parameters. After adding the auxiliary loss used by MoE, the performance decreased.

| Training Loss | ARC-C | ARC-E | BoolQ | HellaSwag | PIQA | RACE | SIQA | LAMBADA | AVG |
|---|---|---|---|---|---|---|---|---|---|
| LM loss only (ours) | 22.4 | 48.6 | 60.3 | 32.7 | 68.3 | 30.9 | 38.6 | 33.3 | **41.9** |
| LM loss + load balance loss | 21.2 | 50.8 | 60.2 | 32.4 | 66.5 | 31.5 | 37.7 | 33.1 | 41.7 |
| LM loss + load balance loss + $z$-loss | 20.7 | 50.5 | 51.7 | 32.5 | 67.7 | 30.8 | 38.2 | 32.5 | 40.6 |

2019), and LAMBADA (Paperno et al., 2016). For all these benchmarks, we report the zero-shot accuracy.

**Offloading Setting.** To measure the deployment efficiency of different models in VRAM-constrained environments, we apply offloading strategies to both MoE and MoLE to ensure their VRAM usage is consistent with that of a dense model with the same number of activated parameters. For the MoE model, we adopt an expert offloading strategy, where only the parameters of the activated experts and all non-expert parameters are stored in VRAM. During each inference step, if the required activated expert is not in VRAM, it is loaded from the storage device into VRAM, replacing the previously loaded expert. For MoLE, we offload the LUT, while keeping the other parameters stored in VRAM.

### 4.2. Main Results

As shown in Table 3, both MoE and MoLE significantly improve performance over the dense baseline. In the com-

parison of five pairs of MoLE and MoE models with the same number of training parameters, MoLE outperforms MoE in four out of the five comparisons in terms of average accuracy. Notably, for MoLE-16E with 160M, 410M, and 1B activated parameters, the number of per-token loaded parameter is only about 1/1500, 1/2000, and 1/2000 of that of MoE, respectively. This demonstrates that MoLE can maintain outstanding performance while significantly reducing communication overhead, making it feasible to offload to lower-tier storage devices.

We note that the size of the LUTs in MoLE is 2.4 to 7.4 times larger than the size of the offloaded experts in MoE. However, since these parameters are offloaded to large, scalable storage devices, we believe the storage overhead of the LUTs remains within an acceptable range. Specifically, as the model size increases, the proportion of the LUTs also decreases accordingly. On models with 1B activated parameters, the size of the LUTs in MoLE-4E becomes

*Table 5.* **Ablation study on the hidden dimension of routed experts.** The base model is MoLE-16E with 160M activated parameters.

| $D_r$ | # Param in Training | # Param Offloaded | # Param Loaded per Token | ARC-C | ARC-E | BoolQ | HellaSwag | PIQA | RACE | SIQA | LAMBADA | AVG |
|---|---|---|---|---|---|---|---|---|---|---|---|---|
| 768 | 0.4B | 7.4B | 0.15M | 20.6 | 48.1 | 58.7 | 31.4 | 66.9 | 30.3 | 38.7 | 31.9 | 40.8 |
| 3072 | 1.1B | 7.4B | 0.15M | 22.4 | 48.6 | 60.3 | 32.7 | 68.3 | 30.9 | 38.6 | 33.3 | 41.9 |
| 12288 | 3.8B | 7.4B | 0.15M | 21.8 | 52.4 | 58.1 | 33.1 | 67.6 | 29.2 | 38.7 | 32.3 | 41.7 |

*Table 6.* **Ablation study on the number of routed experts.** The base model is MoLE with 160M activated parameters.

| $N$ | # Param in Training | # Param Offloaded | # Param Loaded per Token | ARC-C | ARC-E | BoolQ | HellaSwag | PIQA | RACE | SIQA | LAMBADA | AVG |
|---|---|---|---|---|---|---|---|---|---|---|---|---|
| 2 | 0.3B | 0.9B | 0.02M | 19.5 | 48.2 | 57.7 | 30.2 | 64.7 | 29.5 | 38.1 | 29.4 | 39.7 |
| 4 | 0.4B | 1.8B | 0.04M | 21.9 | 48.5 | 60.7 | 31.2 | 65.1 | 29.4 | 38.4 | 31.1 | 40.8 |
| 8 | 0.6B | 3.7B | 0.07M | 19.4 | 50.5 | 60.2 | 32.0 | 66.4 | 29.9 | 37.4 | 32.6 | 41.1 |
| 16 | 1.1B | 7.4B | 0.15M | 22.4 | 48.6 | 60.3 | 32.7 | 68.3 | 30.9 | 38.6 | 33.3 | 41.9 |
| 32 | 2.0B | 14.7B | 0.29M | 21.8 | 53.1 | 59.0 | 33.5 | 68.4 | 30.7 | 38.8 | 33.0 | 42.3 |

comparable to the size of the experts in MoE.

## 4.3. Ablation Experiments

**Training loss.** Unlike MoE, MoLE is a fully differentiable model, so during training, we do not encounter issues like router collapse or instability. Therefore, we do not use any additional auxiliary losses. To illustrate this, we attempt to add the load balance loss and $z$-loss from MoE. As shown in Table 4, after adding these losses, the model's performance decline. This is because the additional losses caused the model's optimization objectives to become misaligned with the inference requirements, leading to negative effects.

**Number and size of experts.** We experiment with varying the size and number of experts. As shown in Table 5, when the hidden dimension of the experts increases from $d$ to $4d$, the model's performance improves. However, further increasing the dimension to $16d$ leads to performance saturation. This is because increasing the size of the experts does not affect the re-parameterized models or the size of the LUTs during inference. It indicates that the knowledge embedded in LUTs with a constant size reaches saturation as the expert size increases, meaning that there is no "free lunch" — further increases in expert size do not lead to additional performance gains.

Unlike the increase in size, the increase in the number of experts results in a continuous performance improvement, demonstrating a certain level of scalability. At the same time, the size of the LUTs and the amount of parameters transferred will also increase proportionally.

**Architecture designs.** To ensure that routed experts can be re-parameterized, we change the input to the routed experts from intermediate features to embedding tokens. Intuitively, this modification means that the experts only receive the raw word features and no context-related information, which is likely to lead to a decrease in model performance. But on the other hand, since the re-parameterized experts do not require computation during the inference phase, we can activate all the experts while keeping the model's inference FLOPs unchanged. This helps compensate for the performance loss mentioned earlier. To conduct an ablation study on this, we trained the following model variants, evolving from MoE-10E to MoLE-4E of 160M activated parameters:

- *Full activation*. All experts of MoE-10E are activated, which causes the number of activated parameters to increase from 0.16B to 0.39B. Meanwhile, all auxiliary losses are discarded.

- *Reconfiguration*. Based on the above model, we modify the experts by changing from 10 routed experts to 1 shared expert and 4 routed experts. Additionally, the hidden dimension of the experts is increased to twice its original size. The total number of parameters remains unchanged.

- *Embedding as inputs*. Based on the above model, we change the input to the routed experts to embedding tokens.

- *Re-parameterization*. Based on the above model, we reparameterize the routed experts as LUTs. Then the

*Table 7.* **Ablation study on the designs in structure.**

| Model | # Param in Training | # Param Activated in Inference | ARC-C | ARC-E | BoolQ | HellaSwag | PIQA | RACE | SIQA | LAMBADA | AVG |
|---|---|---|---|---|---|---|---|---|---|---|---|
| MoE-10E | 0.39B | 0.16B | 21.7 | 49.5 | 51.6 | 32.0 | 66.8 | 30.6 | 39.1 | 31.0 | 40.3 |
| + Full activation | 0.39B | 0.39B | 21.6 | 50.2 | 58.6 | 33.3 | 66.8 | 30.5 | 39.6 | 33.5 | 41.8 |
| + Reconfiguration | 0.39B | 0.39B | 21.5 | 48.7 | 58.0 | 32.7 | 67.5 | 31.1 | 38.5 | 33.7 | 41.5 |
| + Embedding as inputs | 0.39B | 0.39B | 21.9 | 48.5 | 60.7 | 31.2 | 65.1 | 29.4 | 38.4 | 31.1 | 40.8 |
| + Re-param. = MoLE-4E | 0.39B | 0.16B | 21.9 | 48.5 | 60.7 | 31.2 | 65.1 | 29.4 | 38.4 | 31.1 | 40.8 |

*Table 8.* **Post-training quantization for LUTs.** The base model is MoLE-4E with 160M activated parameters.

| LUT Precision | Size of Param Offloaded | Size of Param Loaded per Second | ARC-C | ARC-E | BoolQ | HellaSwag | PIQA | RACE | SIQA | LAMBADA | AVG |
|---|---|---|---|---|---|---|---|---|---|---|---|
| FP16 | 3.5GB | 72KB | 21.9 | 48.5 | 60.7 | 31.2 | 65.1 | 29.4 | 38.4 | 31.1 | 40.8 |
| NF4 | 0.9GB | 18KB | 21.5 | 48.5 | 61.7 | 31.3 | 64.7 | 29.6 | 38.6 | 30.9 | 40.9 |
| NF3 | 0.7GB | 14KB | 22.3 | 48.1 | 59.8 | 31.0 | 65.3 | 28.9 | 38.5 | 30.1 | 40.5 |

number of activated parameters during inference returns to 0.16B. This model is referred to as MoLE-4E.

As shown in the table, using embedding tokens as the input to the routed experts only results in a 0.7 performance drop, but it brings significant benefits of enabling the experts to be re-parameterized, allowing us to activate all experts. A fully activated MoE yields a 1.5 performance gain compared to top-2 MoE, which leads to an overall performance improvement for MoLE over MoE.

### 4.4. Efficiency

We measure the per-step decoding latency of models with 410M activated parameters on NVIDIA V100 GPU using Huggingface's `transformers` package. Since the specific speed of parameter loading is largely influenced by the underlying implementation, we estimate the latency of loading parameters based on the maximum PCIe bandwidth of the V100, which is 16GB/s. For the MoE model, when the batch size is 1, the experts activated in the previous decoding step are retained in VRAM. When the batch size is greater than 1, random two of the activated experts from the previous decoding step are retained in VRAM for each layer. If the experts activated in the current step overlap with those in VRAM, they will not be reloaded. Under this setup, the average number of experts loaded per step for batch sizes of 1, 8, and 32 are 1.6, 6.7, and 8.0 for MoE-10E, or 1.9, 12.3, and 27.4 for MoE-34E, respectively. The input length is fixed to 512.

As shown in Figure 3, the latency of MoLE is comparable to that of the dense model, while MoE exhibits significantly

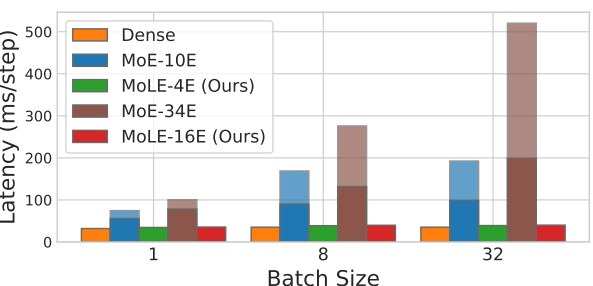

*Figure 3.* **Decoding latency.** We use experts offloading for MoE. The light-colored portion of the bars represents the delay caused by loading.

higher latency than the dense model. As the batch size increases, the number of experts being loaded also increases, further adding to the latency of MoE, but the latency of MoLE has almost no increase.

### 4.5. Reducing the Size of LUTs

Although MoLE significantly reduces data transfer in offloading scenarios compared to MoE, it has a larger storage footprint. While storage space may not be as constrained as VRAM, reducing the size of the LUTs can still alleviate deployment burdens. To address this, we conduct a simple experiment to compress the LUTs. We apply post-training quantization to the FP16 LUTs, quantizing them to NF4 and NF3 (Dettmers et al., 2023) data types. The token-wise block sizes for quantization are 768 and 128,

respectively. As shown in Table 8, the model's performance suffers minimal loss, while the storage burden and size of transferred data are reduced to 25.3% and 19.5% of the original size, respectively. This indicates that the LUTs still contains significant redundancy and has the potential for further compression. We leave this as future work.

## 5. Conclusion

In this paper, we address the issues of high memory consumption and loading latency in MoE by proposing MoLE, a novel language model architecture. MoLE restricts the input to experts to a limited set (embedding tokens), allowing the experts to be re-parameterized into LUTs before inference, thus avoiding the need to load expert parameters. MoLE demonstrates competitive results on downstream tasks, while significantly outperforming MoE with expert offloading in terms of inference speed. This work provides a new direction for designing edge-friendly language models. Future research could explore more diverse discrete spaces and expert architectures.

## Impact Statement

This paper advances LLMs, which have potential societal impacts, including concerns around bias, misinformation, and accessibility. Continued ethical oversight and interdisciplinary collaboration are essential as the field evolves.

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

# Appendix

# A. Pseudocode

### A.1. Training Phase

```python
class MoleDecoderLayer(nn.Module):
    def __init__(self, config):
        super().__init__()
        self.self_attn = Attention(config)
        self.shared_expert = MLP(config)
        self.router = nn.Linear(config.hidden_size, config.num_experts, bias=False)
        self.routed_expert = nn.ModuleList([MLP(config) for _ in config.num_experts])
        self.input_layernorm = RMSNorm(config.hidden_size)
        self.post_attention_layernorm = RMSNorm(config.hidden_size)
        self.expert_layernorm = RMSNorm(config.hidden_size)

    def forward(self, hidden_states, embedding_states):
        '''Attention'''
        residual = hidden_states
        hidden_states = self.input_layernorm(hidden_states)
        hidden_states = self.self_attn(hidden_states)
        hidden_states = residual + hidden_states

        '''Shared Expert'''
        residual = hidden_states
        hidden_states = self.post_attention_layernorm(hidden_states)
        shared_output = self.shared_expert(hidden_states)

        '''Routed Expert'''
        router_value = nn.functional.softmax(self.router(hidden_states), dim=-1)
        embedding_states = self.expert_layernorm(embedding_states)
        routed_output = torch.stack([expert(embedding_states) for expert in
            self.routed_expert], dim=2)
        routed_output = (routed_output * router_value.unsqueeze(-1)).sum(dim=2)
        hidden_states = residual + shared_output + routed_output

        return hidden_states
```

### A.2. Inference Phase

```python
class MoleDecoderLayer(nn.Module):
    def __init__(self, config):
        super().__init__()
        self.self_attn = Attention(config)
        self.shared_expert = MLP(config)
        self.router = nn.Linear(config.hidden_size, config.num_experts, bias=False)
        self.lut = LookupTable(config.vocab_size, config.num_experts * config.hidden_size)
        self.input_layernorm = RMSNorm(config.hidden_size)
        self.post_attention_layernorm = RMSNorm(config.hidden_size)

    def forward(self, hidden_states, input_ids):
        '''Lookup'''
        lookup_results = self.lut(input_ids).to(hidden_states.device, non_blocking=True)

        '''Attention'''
        residual = hidden_states
        hidden_states = self.input_layernorm(hidden_states)
        hidden_states = self.self_attn(hidden_states)
        hidden_states = residual + hidden_states

        '''Shared Expert'''
        residual = hidden_states
```

```
    hidden_states = self.post_attention_layernorm(hidden_states)
    shared_output = self.shared_expert(hidden_states)

    '''Routed Expert'''
    router_value = nn.functional.softmax(self.router(hidden_states), dim=-1)
    lookup_results = lookup_results.view(-1, config.num_experts, config.hidden_size)
    routed_output = (lookup_results * router_value.unsqueeze(-1)).sum(dim=2)
    hidden_states = residual + shared_output + routed_output

    return hidden_states
```

## B. Hyper-parameters

| Configuration Key | Value |
|---|---|
| attention-dropout | 0 |
| dtype | bf16 |
| global-batch-size | 1024 |
| gradient-clipping | 1.0 |
| hidden-dropout | 0 |
| lr-decay-style | cosine |
| max-position-embeddings | 2048 |
| min-lr | $0.1 * optimizer.params.lr$ |
| no-weight-tying | True |
| norm | RMSNorm |
| optimizer.params.betas | [0.9, 0.95] |
| optimizer.params.eps | 1e-08 |
| optimizer.type | Adam |
| pos-emb | rotary |
| rotary-pct | 0.25 |
| seq-length | 2048 |
| train-iters | 50000 |
| warmup | 0.01 |
| weight-decay | 0.01 |

