# OpenReview forum: "Mixture of Lookup Experts"
_ICML.cc/2025/Conference — ICML 2025 oral_

### Official Review · Reviewer_VEVw · 2025-03-07

**Overall Recommendation:** 4

**Summary:**

This paper introduces a new LLM architecture, MoLoE. MoLoE enables the conversion of experts into lookup tables before inference, which does not require computation and is placed outside of VRAM, thereby reducing its VRAM usage to levels comparable to dense models. At the same time, MoLoE only needs to transfer the lookup results into VRAM during inference, without the need to transfer any model parameters, making its extra latency for data transfer negligible. This addresses the issues of high VRAM usage in MoE models and the significant latency caused by expert offloading. Experiments on a range of language understanding datasets demonstrate that MoLoE can achieve performance comparable to, or even better than, MoE models.


## Update After Rebuttal

The authors’ response has adequately addressed my key concerns, particularly regarding the long-context optimization strategies and MoE’s inefficiency in single-GPU deployment. I am keeping my rating.

**Claims And Evidence:**

The paper claims that MoLoE achieves lower latency and better performance compared to MoE, with the same VRAM usage and number of activated parameters. These claims are supported by experimental results.

**Essential References Not Discussed:**

As mentioned above, due to the similar objectives, expert pruning methods should be discussed, such as [1].

[1] Not All Experts are Equal: Efficient Expert Pruning and Skipping for Mixture-of-Experts Large Language Models, ACL 2024.

**Experimental Designs Or Analyses:**

The paper evaluates the model on language understanding tasks and compares it with dense and MoE models. The comparison is conducted under the control of the same number of activated parameters and VRAM usage, making it convincing.

**Methods And Evaluation Criteria:**

This paper essentially reduces VRAM usage and inference latency at the cost of increased storage overhead outside of VRAM. For scenarios with limited VRAM and sensitivity to latency (such as edge-side large language models), MoLoE holds practical value.

**Other Comments Or Suggestions:**

See above.

**Other Strengths And Weaknesses:**

**Strengths:**

1. The motivation is clear, and the method makes sense. This paper introduces a novel approach for reducing the VRAM usage of MoE.
2. The paper demonstrates the effectiveness of MoLoE through comprehensive experiments.

**Weaknesses:**

1. In certain settings, MoLoE’s offloaded parameters are significantly more than those of MoE. For example, in the setting with 160M activated parameters, the offloaded parameters of MoLoE-16E are more than seven times those of MoE. As the model scales up, the usability of MoLoE may come into question.
2. Theoretically, when the prompt length is sufficiently long, the lookup results that MoLoE needs to load during pre-filling could surpass the size of MoE's experts. For example, in a model with 1B activated parameters, the MoE-10E expert parameter count is 2.7B (assuming all experts are loaded for each layer and then deleted after computation). On the other hand, MoLoE-4E loads 0.26M parameters per token. This means that when the prompt length exceeds 10k, MoLoE-4E would need to load more parameters than MoE-10E.

**Questions For Authors:**

1. In Figure 3, why is MoE significantly slower than the dense model, even when ignoring the loading time, despite both having the same number of activated parameters?
2. Since the experts in MoLoE can be reparameterized as LUTs, the specific architecture of the experts during training should not affect the architecture during inference. Is it possible to use modules other than FFN as experts?

**Relation To Broader Scientific Literature:**

Since this paper aims to address the issue of large parameter sizes in MoE, it seems to align with the goal of expert pruning.

**Theoretical Claims:**

This paper is application-oriented and does not involve theoretical proofs. The explanatory formulas and parameters/FLOP calculations presented in the paper are correct.

---

> ### Author Rebuttal · Authors · 2025-03-31
>
> We thank the reviewer for their thoughtful feedback. We address specific concerns and questions below.
>
> > Q1. As the model scales up, the usability of MoLoE may come into question.
>
> Firstly, the number of LUT parameters that need to be offloaded by our method is $dN|V|$, whereas MoE needs to offload the expert size of $2dND_r$. Since $|V|$ is typically on the order of tens of thousands, the difference between the two is not greater than an order of magnitude. Additionally, the method is designed for edge deployment of LLMs, where the parameter size is constrained by the available computational resources, typically resulting in smaller models. Therefore, the size of the LUTs stored on lower-level storage devices is generally acceptable.
>
> > Q2. When the prompt length exceeds 10k, MoLoE-4E would need to load more parameters than MoE-10E
>
> In the extreme case of very long texts, the amount of parameters our method loads during the pre-filling phase may indeed be greater than that of MoE. However, during LLM inference, the time required for multi-step decoding remains the dominant factor, meaning the cost during the prefilling phase can be somewhat offset. Additionally, directly loading the expert parameters before reparameterization during the pre-filling phase and using computation to replace loading is another potential solution.
>
> > Q3. Why is MoE significantly slower than the dense model?
>
> Our implementation is based on the Mixtral code from the Huggingface transformers package. Unlike cluster environments that can utilize expert parallelism, edge-side single-card deployment requires each activated expert to process tokens individually, resulting in lower parallelism. When the batch size is greater than 1, the parallelism is further reduced due to different tokens requiring different experts, leading to slower speeds. It is worth noting that the inefficiency of MoE computation could potentially be addressed in the future through more friendly designs of hardwares or better operator implementations. However, even if the single-card execution efficiency of MoE becomes comparable to that of dense models, our method still maintains a significant advantage in terms of the latency associated with loading parameters.
>
> > Q4. Is it possible to use modules other than FFN as experts?
>
> In fact, as long as the module does not involve information exchange between tokens, i.e., the module’s input is context-independent, it can serve as an expert in MoLoE. This characteristic is similar to the FFN layer in transformers. Therefore, we adopt FFN as the expert, ensuring that the type of expert is consistent with the MoE model.

---

### Official Review · Reviewer_vBLo · 2025-03-09

**Overall Recommendation:** 4

**Summary:**

This paper presents MoLoE, a new MoE architecture designed to address the high VRAM usage of traditional MoE models. MoLoE uses the output of the Embedding layer as input for the experts during training. Before inference, it pre-computes the output of each expert for each token ID, reparameterizing the experts as LUTs. By offloading the LUTs to other storage devices, MoLoE reduces the memory overhead of the experts while introducing negligible data transfer latency, and achieves performance superior to MoE with the same number of activated parameters.

## update after rebuttal

Thanks for addressing my questions, I keep my rating as accept.

**Claims And Evidence:**

The paper primarily makes the following claims:
1. MoE faces deployment challenges, including high VRAM usage or significant offload latency.
2. MoLoE can reparameterize the experts during the inference stage, resulting in lower VRAM usage and inference latency.
3. MoLoE achieves performance comparable to or better than MoE.

These claims are either theoretically valid or can be empirically demonstrated through experiments.

**Essential References Not Discussed:**

Although it can be considered as concurrent work, the authors are encouraged to include the latest papers from ICLR 2025 in the related work section.

+ MoE++: Accelerating Mixture-of-Experts Methods with Zero-Computation Experts
+ TC-MoE: Augmenting Mixture of Experts with Ternary Expert Choice

**Experimental Designs Or Analyses:**

The experiments are conducted on 8 tasks, with dense and MoE models used as baselines for comparison. The metrics include accuracy, number of activated parameters, number of parameters during training, offloaded parameters, and number of data transferred. The experiments demonstrate the validity and effectiveness of the proposed method.

**Methods And Evaluation Criteria:**

The proposed method reduces data transfer by several orders of magnitude compared to MoE, without sacrificing performance, while maintaining the same number of activated parameters and VRAM usage. This aligns with the objectives of the paper. However, a side effect of this approach is the need to maintain large LUTs in external storage devices, which may limit the applicability of the model in certain scenarios.

**Other Comments Or Suggestions:**

+ Add citations to the papers mentioned above.

**Other Strengths And Weaknesses:**

Strengths:
+ MoLoE addresses the challenges of MoE deployment -- high VRAM requirements and the large delays associated with offloading experts, offering practical value.
+ The experiments demonstrate that the model achieves better performance than MoE while reducing the number of transferred parameters by over a thousand times.

Weaknesses:
+ As a trade-off for the strengths mentioned above, the model needs to store larger LUTs on storage devices, potentially several times larger than those of MoE, which may limit the applicability of the model in certain scenarios.

**Questions For Authors:**

+ Are there other ways to reduce the size of the LUT, such as pruning LUT entries corresponding to rare token IDs?

+ Why did the MoE baselines not use shared experts?

**Relation To Broader Scientific Literature:**

The method seems to be related to reparameterizable network architectures in computer vision, as both approaches simplify the model by reparameterizing after training. For example:

+ RepVGG: Making VGG-style ConvNets Great Again

**Theoretical Claims:**

The paper does not introduce particularly complex techniques, and its effectiveness is primarily demonstrated through experimental results. The ablation study in Table 7 effectively highlights the validity of the model design.

---

> ### Author Rebuttal · Authors · 2025-03-31
>
> We thank the reviewer for their thoughtful feedback. We address specific concerns and questions below.
>
> > Q1. Pruning LUT entries corresponding to rare token IDs.
>
> We prune half entries of the LUTs of MoLoE-16E with 160M activated parameters based on frequency. When the input token ID is pruned, the LUTs return a zero vector. The results are as follows:
>
> |Model|#offloaded param|#activated param|ARC-C|ARC-E|BoolQ|HellaSwag|PIQA|RACE|SIQA|LAMBADA|average|
> |-|-|-|-|-|-|-|-|-|-|-|-|
> |MoLoE-16E|7.4B|160M|22.4|48.6|60.3|32.7|68.3|30.9|38.6|33.3|41.9|
> |MoLoE-16E-pruned|3.7B|160M|21.7|44.8|61.8|31.2|64.5|28.8|39.3|26.6|39.8|
>
> From the experimental results, it is evident that post-training LUT pruning significantly degrades model performance. If smaller LUTs are required, using a more compact vocabulary during the training phase is a better choice.
>
> > Q2. Why did the MoE baselines not use shared experts?
>
> The different settings for MoE and MoLoE are chosen in order to prioritize aligning both **total parameter count in training** and **activated parameter count in inference** for a fair comparison.
>
> + MoLoE adopts shared experts because, unlike MoE, MoLoE reparameterizes all routed experts into LUTs during inference. If shared experts were not used, MoLoE would not include any FFN layers during inference, which would make it difficult to align its activated parameter count (VRAM usage) with that of dense and MoE models.
> + The reason MoE does not adopt shared experts is that, in our experiments, MoE without shared experts performs better, which has also been thoroughly validated in [1]. We provide our experimental results below (No.1 vs No.2).
> + In inference stage, MoLoE only has 1 shared expert, while MoE activates 2 experts. Therefore, to make a fair comparison, we set the size of each expert in MoE to half of that in MoLoE (to align activated parameter count in inference), but double the number of experts (to align total parameter count in training). As a result, the total number of experts cannot be aligned (No.2 vs No.3). But as a reference, we have still provided the results below for MoLoE and MoE both using 1 shared expert and 33 routed experts (No.2 vs No.4).
>
> |No.|Model|#total param in training|#activated param|ARC-C|ARC-E|BoolQ|HellaSwag|PIQA|RACE|SIQA|LAMBADA|average|
> |-|-|-|-|-|-|-|-|-|-|-|-|-|
> |1|MoE (0 shared, top-2 from 34 routed)|1.1B|160M|20.5|50.0|57.5|34.5|67.3|28.6|39.9|36.4|41.8|
> |2|MoE (1 shared, top-1 from 33 routed)|1.1B|160M|20.6|48.4|57.3|31.5|65.2|30.5|37.9|34.7|40.8|
> |3|MoLoE (1 shared, 16 routed)|1.1B|160M|22.4|48.6|60.3|32.7|68.3|30.9|38.6|33.3|41.9|
> |4|MoLoE (1 shared, 33 routed)|2.1B|160M|22.0|52.4|59.3|33.5|68.3|30.7|38.8|33.1|42.2|
>
> [1] OLMoE: Open Mixture-of-Experts Language Models. ICLR 2025
>
> > Q3. Add citations to the papers mentioned above.
>
> Thank you for your suggestion, we will update it in the revised version.

---

### Official Review · Reviewer_xkaR · 2025-03-11

**Overall Recommendation:** 4

**Summary:**

This paper proposes MoLoE architecture to address the high memory overhead of MoE architectures. The key difference is that MoLoE converts experts into external LUTs before inference, eliminating expert computation and allowing experts to be stored outside of VRAM. Additionally, since only the lookup results need to be transferred to the GPU, MoLoE nearly eliminates the parameter transmission overhead associated with offloading. Experimental results demonstrate that MoLoE achieves competitive performance with MoE while maintaining latency and VRAM usage comparable to dense models.

**Claims And Evidence:**

The claims made in the paper are supported by experiments.

**Essential References Not Discussed:**

N/A

**Experimental Designs Or Analyses:**

The experimental section includes comparisons with MoE and dense models, as well as ablation studies, which are sound and valid.

**Methods And Evaluation Criteria:**

MoLoE meets the requirements of the intended application scenarios. The benchmarks used in the paper are commonly adopted for LLM evaluation.

**Other Comments Or Suggestions:**

N/A

**Other Strengths And Weaknesses:**

Pros:
1. This paper focuses on the challenges MoE faces during deployment, and MoLoE offers a novel and practical solution.
2. The writing is clear.
3. The experiments are thorough and convincing.

Cons:
1. MoLoE has more offloaded parameters than MoE. Although off-device storage is larger, this still requires additional deployment overhead.
2. MoLoE activates all expert parameters during each training step, which means that during training, the FLOPs of MoLoE will be higher than that of MoE with the same number of parameters.

**Questions For Authors:**

1. When the number of parameters is the same during the training phase, will MoLoE require more training time than MoE?
2. Intuitively, using embeddings as expert inputs would limit the experts to only accepting context-independent, shallow features. So, why is MoLoE able to perform better than MoE?

**Relation To Broader Scientific Literature:**

MoLoE discretizes inputs into a finite set by using embeddings as expert inputs, which is conceptually related to quantization methods. Input quantization could be a potential alternative solution for MoLoE.

**Theoretical Claims:**

No theoretical proof is presented.

---

> ### Author Rebuttal · Authors · 2025-03-31
>
> We thank the reviewer for their thoughtful feedback. We address specific concerns and questions below.
>
>
>
> > Q1. Will MoLoE require more training time than MoE?
>
> Since MoLoE activates all parameters during training, its training cost is approximately equal to that of a dense model with the same number of parameters, and higher than that of a sparse MoE model with the same parameter count. However, it should be noted that our method is designed for edge deployment of LLMs, where the parameter size is limited by the computational resources available on the edge, typically resulting in smaller models. Therefore, the cost of training these smaller-scale LLMs on a cluster remains manageable.
>
> > Q2. Why is MoLoE able to perform better than MoE?
>
> + Although the experts' inputs do not contain context information, they can still indirectly influence the model's handling of context by altering the input to subsequent attention layers. Additionally, the retained shared expert operates similarly to MoE and dense models, where it takes the output from the previous layer as its input, and thus can also handle context-related information.
> + By changing the expert inputs to embedding tokens, the routed experts can be reparameterized as LUTs, resulting in almost zero computation during inference. This allows us to activate all experts without incurring additional costs, which helps recover some of the performance loss caused by altering the expert inputs (see Table 7). Additionally, since activating all experts does not require auxiliary losses, the training and inference objectives are more aligned, leading to improved model performance (see Table 4).

---

### Official Review · Reviewer_iLJn · 2025-03-14

**Overall Recommendation:** 3

**Summary:**

The paper proposes Mixture of Lookup Experts (MoLoE), a new variation of Mixture-of-Experts (MoE) architectures that significantly reduces GPU inference latency for batched generation.

**Claims And Evidence:**

I think the claims and evidence are good.

**Essential References Not Discussed:**

I think it's better to compared with Hwang, Ranggi, et al. "Pre-gated moe: An algorithm-system co-design for fast and scalable mixture-of-expert inference." 2024 ACM/IEEE 51st Annual International Symposium on Computer Architecture (ISCA). IEEE, 2024.

**Experimental Designs Or Analyses:**

I checked the experimental designs and analyses. Overall, it is good, but it needs more alignment and baselines.

**Methods And Evaluation Criteria:**

Something is not aligned with the baselines.

**Other Comments Or Suggestions:**

I think some system-level optimization work should be discussed or as a baseline.
e.g. Hwang, Ranggi, et al. "Pre-gated moe: An algorithm-system co-design for fast and scalable mixture-of-expert inference." 2024 ACM/IEEE 51st Annual International Symposium on Computer Architecture (ISCA). IEEE, 2024.

**Other Strengths And Weaknesses:**

**Strengths:**
1. Introduce the LUT to combine the advantages of the Dense Model (Balance Training) and MoE architecture (Parameter Scaling).
2. Training all experts simultaneously removes the need for auxiliary losses, simplifying training and improving efficiency.

**Weaknesses:**
1. In Table 3, the comparison between MoE (0 shared, top-2 routing) and MoLoE (1 shared, 4 routed) is inconsistent. I think a direct comparison under identical conditions would strengthen the evaluation.
2. I have concerns about the context information in your MoLoE model. After being re-parameterized, during the inference, the input ID and output are fixed for each token. As Table 3 shows, performance under LAMBADA is weaker than MoE. Could you provide results under complex language modeling tasks like WikiText or Winogrande?
3. As I understand, the tokenizer is important for the LUT's construction. Have different tokenizer settings been explored? An ablation study or discussion regarding tokenizer will be better.

**Questions For Authors:**

See Weaknesses.

**Relation To Broader Scientific Literature:**

This work mainly focuses on the loading latency for MoE models in the offloading scenario.

**Theoretical Claims:**

There are no theoretical claims in this paper.

---

> ### Author Rebuttal · Authors · 2025-03-31
>
> We thank the reviewer for their thoughtful feedback. We address specific concerns and questions below.
>
>
> > Q1. The comparison between MoE and MoLoE is inconsistent.
>
> The different settings for MoE and MoLoE are chosen in order to prioritize aligning both **total parameter count in training** and **activated parameter count in inference** for a fair comparison.
>
> + MoLoE adopts shared experts because, unlike MoE, MoLoE reparameterizes all routed experts into LUTs during inference. If shared experts were not used, MoLoE would not include any FFN layers during inference, which would make it difficult to align its activated parameter count (VRAM usage) with that of dense and MoE models.
> + The reason MoE does not adopt shared experts is that, in our experiments, MoE without shared experts performs better, which has also been thoroughly validated in [1]. We provide our experimental results below (No.1 vs No.2).
> + In inference stage, MoLoE only has 1 shared expert, while MoE activates 2 experts. Therefore, to make a fair comparison, we set the size of each expert in MoE to half of that in MoLoE (to align activated parameter count in inference), but double the number of experts (to align total parameter count in training). As a result, the total number of experts cannot be aligned (No.2 vs No.3). But as a reference, we have still provided the results below for MoLoE and MoE both using 1 shared expert and 33 routed experts (No.2 vs No.4).
>
> |No.|Model|#total param in training|#activated param in inference|ARC-C|ARC-E|BoolQ|HellaSwag|PIQA|RACE|SIQA|LAMBADA|average|
> |-|-|-|-|-|-|-|-|-|-|-|-|-|
> |1|MoE (0 shared, top-2 from 34 routed)|1.1B|160M|20.5|50.0|57.5|34.5|67.3|28.6|39.9|36.4|41.8|
> |2|MoE (1 shared, top-1 from 33 routed)|1.1B|160M|20.6|48.4|57.3|31.5|65.2|30.5|37.9|34.7|40.8|
> |3|MoLoE (1 shared, 16 routed)|1.1B|160M|22.4|48.6|60.3|32.7|68.3|30.9|38.6|33.3|41.9|
> |4|MoLoE (1 shared, 33 routed)|2.1B|160M|22.0|52.4|59.3|33.5|68.3|30.7|38.8|33.1|42.2|
>
> [1] OLMoE: Open Mixture-of-Experts Language Models. ICLR 2025
>
> > Q2. Results under complex language modeling tasks like WikiText or Winogrande.
>
> We reported the results for a 1B activated parameter model on WikiText and Winogrande. MoLoE outperforms MoE on Winogrande, but performs slightly worse on WikiText.
>
> |Model|#Param Loaded per Token|WikiText (ppl↓)|Winogrande (acc↑)|
> |-|-|-|-|
> |Dense|0|17.7|54.0|
> |MoE-10E|537M|15.6|55.3|
> |MoLoE-4E|0.26M|16.2|56.5|
>
> Although the routed experts in MoLoE do not directly receive context information, they can still indirectly influence the model's handling of context by altering the input to subsequent attention layers. Additionally, the retained shared expert operates similarly to MoE and dense models, taking the output from the previous layer as its input, and thus can also handle context-related information.
>
> > Q3. Different tokenizer settings.
>
> In addition to Pythia's tokenizer with a vocabulary size of 50k that we used, we also tried the 32k vocabulary tokenizer from Mixtral. The results are shown below.
>
> |Model|#offloaded param|#activated param|#Param Loaded per Token|8-task average|
> |-|-|-|-|-|
> |MoE-10E (Pythia tokenizer)|0.3B|160M|57M|40.3|
> |MoLoE-4E (Pythia tokenizer)|1.8B|160M|0.037M|40.8|
> |MoE-10E (Mixtral tokenizer)|0.3B|134M|57M|39.8|
> |MoLoE-4E (Mixtral tokenizer)|1.2B|134M|0.037M|40.2|
>
> The primary impact of the tokenizer on MoLoE is that its vocabulary size is proportional to the size of the LUTs. However, since different tokenizers also affect the performance of all kinds of models, it is difficult to isolate the tokenizer's specific effect on MoLoE's performance. Nonetheless, at least from the experimental results, we can observe that MoLoE is effective under different tokenizers.
>
> > Q4. Compared with Pre-gated MoE.
>
> Pre-gated MoE allows for the parallelization of expert prefetching and computation. Compared to this approach, our method has the following advantages:
> + Pre-gated MoE does not increase inference latency under the assumption that the time to load parameters is less than or equal to the computation time. However, during batch decoding, since different tokens require different experts, the number of experts to prefetch increases (in the worst case, all experts need to be loaded), which in turn increases the latency.
> + Additionally, when deployed on edge devices, it may be necessary to offload to lower-tier storage devices such as SSDs, where the transfer bandwidth is much smaller, causing the time to load parameters to be significantly higher than computation time, leading to a substantial increase in overall latency.
> + In contrast, our MoLoE fundamentally reduces the amount of data that needs to be transferred by more than three orders of magnitude, thus avoiding additional latency during decoding. The two methods actually offer two different solutions in distinct directions.

---

### Decision · Program_Chairs · 2025-05-01

**Decision:**

Accept (oral)

**Comment:**

This submission worked on mixture of experts (MoE) and proposed mixture of lookup experts (MoLoE), a new MoE architecture that is efficient in both communication and VRAM usage, where the experts can be re-parameterized as lookup tables that retrieves expert outputs based on input IDs and offloaded to storage devices. The authors did a particularly good job in their rebuttal to address most of the concerns raised by the four reviewers. More specifically, three reviewers gave the accept; the only negative reviewer whose overall recommendation is still the weak reject, Reviewer iLJn, did not argue against novelty or practicality, and his or her requirements for more experiments were satisfied by the authors (especially the first weakness). I think the paper is potentially very impactful in the near future given its important topic and nice idea with good performance so that it may significantly push forward the research and/or applicability of MoE architectures. Therefore, we should definitely accept this submission to be published at ICML 2025.